# Road properties of cement-phosphogypsum-red clay under dry and wet cycles

**Yunke Liu, Kaisheng Chen** *, **Hao Jian, Zeyu Liu**

School of Civil Engineering, Guizhou University, Guiyang, Guizhou Province, China

* chen_kaisheng@163.com

**Data Availability Statement:** All relevant data are within the article and its Supporting Information files.

## Abstract

In this paper, the road performance and mechanism of cement-phosphogypsum-red clay (CPRC) under dry and wet cycling were systematically investigated using 5% cement as curing agent, the mass ratio of phosphogypsum: red clay = 1:1, and 5% SCA-2 as water stabilizer. The road performance of dry and wet cycle mix was verified with the National Highway G210 Duyun Yangan to Yingshan Highway Reconstruction and Expansion Project as a test road to provide a scientific basis for the application of cement-phosphogypsum-red clay on roads. The results show that the cement-phosphogypsum-red clay unconfined compressive strength decreases with the increase of the number of wet and dry cycles, with a larger decay in the first three times and leveling off thereafter. The CBR value meets the requirements of roadbed fill on highway and Class I roads as stipulated in the Design Code for Highway Roadbeds (JTG D30-2015). SCA-2 water stabilizer reduces the strength of the mixture, but significantly improve the water stability performance of the mixture, the reason SCA-2 water stabilizer active ingredients and mixing particles of physicochemical reaction, the content of ettringite in the mixture is lower than the content of the mixture not mixed with water stabilizers, the generation of quartz, white mica, and kaolinite and other hydrophilic poor material increases, the surface of the cementitious material increases, the seepage channel is reduced, so the strength is reduced, and water stability performance is improved. The roadbed of the test road was smooth and dense, with no scouring, peeling or cracking. The settlement of the roadbed is less than the requirement of "Highway Roadbed Design Specification" (JTGD30-2015). Cement-phosphogypsum-red clay (cement 5%, phosphogypsum 47.5%, red clay 47.5%, SCA-2 water stabilizer 5%) can be used as filler for road bed on highway and primary road base.

## Introduction

Red clay is widely distributed in Guizhou Province, China, and has engineering properties such as high water content, high plasticity, and high pore ratio. Due to the poor water stability of red clay, compaction difficulties, easy to dry shrinkage and cracking, as a roadbed fill need to add other materials to improve and increase the cost of the project [1]. Phosphogypsum, as a waste discharged from industry, is mostly disposed of in open piles, which pollutes the living

**Funding:** This work was supported by the Guizhou Provincial Science and Technology Program (Qiankehe Basic-ZK [2023] Key 016) and the Guiyang Municipal Science and Technology Program Project (Zhuke Contract [2024]-1-12).

environment of human beings and even affects their health. At present, the global annual production of phosphogypsum is 100~280 million tons, and the utilization rate of phosphogypsum is only about 15% [2]. Resource utilization of phosphogypsum has always been a worldwide problem, and "disposal and discharge" is usually the main method used to deal with phosphogypsum all over the world. Guizhou has Kaiyang, urn Fu two large phosphate-rich mines, phosphate fertilizers and phosphorus chemical industry has become a pillar industry in Guizhou, the total production capacity has reached more than 1 million t / a, in order to solve the problem of phosphogypsum piling, since August 2023 Guizhou Phosphorus Group has launched the implementation of three batches of scientific and technological innovation, "unveiled to lead the question," the list of projects, the highest reward is 3 million yuan. In order to consume phosphogypsum and realize production by volume, the People's Government of Guizhou Province and the People's Government of Guiyang Municipality have issued a series of documents to accelerate the promotion of phosphogypsum policy measures. The mixing of phosphogypsum and red clay for roadbed engineering can not only solve the problem of phosphogypsum stockpiling, but also improve the engineering properties of red clay, which has certain academic significance and engineering application value.

Numerous scholars have now conducted numerous studies on the mechanical properties of phosphogypsum stabilised soil mixes and the feasibility of using them for road construction. James, J [3] analysed phosphogypsum as an additive to lime to improve its performance in soil stabilisation and found that the addition of phosphogypsum to lime improves the early and late strength of stabilised soils. Liu et al. [4] found that the water stability of the mixture was poor and the unconfined compressive strength tended to increase and then decrease with the increase of phosphogypsum content through the study of lime-phosphogypsum stabilized red clay. Amrani, M et al. [5] found that when phosphogypsum: clay = 40:60, the mix has good resistance to leaching, load bearing capacity and compaction characteristics and when calcareous material: phosphogypsum: clay = 10:25:65 with 7% external road binder, the mix has the best mechanical properties and is capable of being used as a pavement material. Zhang et al. [6] conducted a field test study on the mixture by using phosphogypsum stabilized red clay as roadbed filler and concluded that cemented phosphogypsum red clay can be used as roadbed filler. In a study exploring the use of heavily doped phosphogypsum for roadbed filler, Li et al. [7] found that suitable hydrophobic agents ensured the strength growth and water erosion resistance properties of heavily doped phosphogypsum. Xiao, W [8] used sludge as the road base material, ordinary silicate cement, fly ash and silica fume as the curing agent, the curing agent was added to the phosphogypsum stabilised sludge mixture, the water stability of the improved phosphogypsum stabilised sludge mixture was obtained to meet the requirements of the pavement design and there was a significant increase in the strength of the mixture. Yue et al. [9] modified phosphogypsum by incorporating slag, calcium-rich silica actives and activated powders into it and found that modified phosphogypsum stabilised pulverised soil mixtures with 12% incorporation had the best mechanical properties and good water stability. Tao et al. [10] uses fly ash, water glass and quicklime to improve phosphogypsum, which dramatically improves the water stability and strength properties of phosphogypsum, and can meet the technical requirements of roadbed and even pavement sub-base; Du et al. [11] used different dosage of solid curing agent and liquid curing agent (dosage 0.5%) to improve cement phosphogypsum stabilized materials, studied the water stability properties and mechanical properties of cement phosphogypsum materials, analyzed the strength formation mechanism, and evaluated the feasibility of its use as a pavement base material; Liu et al. [12] prepared a cement-phosphogypsum stabilized crushed stone material by partially replacing fine aggregate with phosphogypsum, and the results showed that the cement dose, phosphogypsum dosage and aggregate grading all had a large effect on the strength of cement-phosphogypsum

**Table 1. Basic physical indexes of red clay.**

| $\rho/\text{g}\cdot\text{cm}^{-3}$ | $\omega/\%$ | $\omega_{op}/\%$ | $\rho_{dmax}/\text{g}\cdot\text{cm}^{-3}$ | $W_L/\%$ | $W_P/\%$ | Cu | Cc |
|---|---|---|---|---|---|---|---|
| 1.76 | 60.03 | 30.24 | 1.46 | 82.13 | 43.02 | 10.63 | 1.085 |

stabilized crushed stone material. Different proportioning schemes were used for the test sections of the He-Tong highway in Shaanxi province, i.e., Lime:Phosphogypsum:Soil = 5:7:88, Lime:Phosphogypsum:Soil = 6:9:85, and Lime:Phosphogypsum:Soil = 7:10:83, respectively. Through the above research, it can be seen that the existing phosphogypsum materials used in the roadbed pavement, limited to the phosphogypsum mixed with other materials (soil, gravel, sand, fly ash), play an auxiliary role rather than as the main part of the use of the utilization rate is low (generally no more than 15%), and can not be a good solution to the problem of phosphogypsum large amounts of accumulation. In this paper, the road performance of cement-phosphogypsum-red clay (CPRC) under dry and wet cycling was investigated by using 5% cement as curing agent with the mass ratio of phosphogypsum: red clay = 1:1, plus 5% SCA-2 as water stabilizer, through the unconfined compressive strength test, the California bearing ratio (CBR) test, and the water stability test. The water stabilization mechanism of the mix was analyzed by Scanning Electron Microscope and X-ray diffraction tests. The road performance of dry and wet cycle mix was verified with the National Highway G210 Duyun Yangan to Yingshan Highway Reconstruction and Expansion Project as a test road to provide a scientific basis for the application of cement-phosphogypsum-red clay on roads. The significance of this is to improve the engineering properties of red clay while also improving the low utilization of phosphogypsum globally, reducing the pollution of phosphogypsum to the environment, and from the economic point of view, saving the cost of cement, providing a theoretical basis for the application of cement-phosphogypsum-red clay in road engineering.

## Properties of raw materials

The red clay soil used in the test was taken from the section K14+000~K16+000 of Fuquan City Reconstruction and Expansion Project in China, the depth of the soil taken was 0~3m, the soil samples were reddish-brown in colour, dense in structure, with high natural water content and high cohesive cohesion, which was a kind of high liquid limit clay. Table 1 presents the basic parameters of the soil samples, while Table 2 illustrates their chemical composition.

Phosphogypsum was taken from urnfu phosphorus mine in Fuquan City, Guizhou Province, with large water content, dark brown and grey. The basic physical index of phosphogypsum is shown in Table 3, the chemical composition is shown in Table 4, and the detection results of heavy metal and radioactivity are shown in Table 5. According to the detection results in Table 5, the content of heavy metal in phosphogypsum does not exceed the relevant provisions of national standard (GB 5085.3–2007), and the radioactivity index does not exceed the relevant provisions of national standard (GB 6566–2010).

**Table 2. Chemical composition of red clay.**

| Si/% | Fe/% | Al/% | K/% | Mg/% | O/% |
|---|---|---|---|---|---|
| 25.84 | 8.02 | 14.75 | 2.68 | 0.88 | 43.56 |
| $SiO_2/\%$ | $Al_2O_3/\%$ | $Fe_2O_3/\%$ | $K_2O/\%$ | MgO/% | $TiO_2/\%$ |
| 55.60 | 27.32 | 11.26 | 3.12 | 1.39 | 1.20 |

**Table 3. Basic parameters of phosphogypsum.**

| Specific surface area / $m^2 \cdot kg^{-1}$ | Loss on ignition /% | Moisture content /% | Alkali content /% | Density /$g \cdot cm^{-3}$ | fineness /% |
|---|---|---|---|---|---|
| 102 | 18.43 | 5.3 | 1.31 | 2.38 | 44.3 |

The cement is Southwest brand P.C32.5R silicate cement, grey, dry, no lumps, the basic parameters are shown in Table 6. After testing, all the indexes of the cement sample are in line with the requirements of the technical indexes of General Silicate Cement (GB 175–2007).

The basic parameters of SCA-2 curing agent are shown in Table 7, respectively.

## Design of mix ratios

According to previous research results [13–20], the maximum unconfined compressive strength of cemented phosphogypsum-stabilised soil was achieved when the cement dosage was 4%-6% and the ratio of cement to phosphogypsum mass was 1:1 to 1:3 [21]. Therefore, the design of the mix ratio can be carried out to effectively improve the strength properties of the mix at the lowest cost.

### Mixing ratio design

When cement stabilised soil is used as sub-base and the plasticity index of the soil is greater than 12, the recommended cement dosage in the Technical Rules for the Construction of Highway Pavement Base Levels (JTG/T F20-2015) [22] is 6%-14%. According to Jinxiong Chen et al. [23] by studying the stabilization of red clay by cement phosphogypsum, it was concluded that the best unconfined compressive strength of the mixture can be obtained when the cement dosage is at 6%, which can meet the requirements of highway roadbed filler; According to the previous research results [24,25], when the cement is 5%, the road performance of the mixture is good and can be used as roadbed filler, too much cement will lead to shrinkage cracks in cement stabilized materials, so the cement dosage is set at 5%. In order to achieve a large amount of phosphogypsum consumption, the study of high dosage of phosphogypsum stabilized red clay road performance, set phosphogypsum: red clay = 1:1, in order to improve the water stability performance of the mixture, according to Ji Xiaoping and Huang Wendong [26,27] on the study of curing agent stabilized phosphogypsum roadbase filler, the curing agent can improve the strength and water stability of the phosphogypsum roadbase filler, so externally doped with 5% SCA-2 type water stabilizer. Highway Roadbed Design Specification (JTG D30-2015) stipulates that the compaction degree of roadbed on the roadbed of first-class highway and motorway is required to reach 96%, so the compaction degree of the mixture is taken as 96%. The specimen mix ratio is shown in Table 8.

### Sample preparation

Specimen preparation according to "highway engineering inorganic binding material stabilizer test procedures" (JTG E51-2009) [28]: need to remove the red clay and phosphogypsum in the particle size of more than 2mm particles, according to the ratio design, the optimal moisture content should be added to the calculation of the amount of water, set aside 3% of

**Table 4. Chemical composition of phosphogypsum.**

| Ingredient | $SO_3$ | CaO | $SiO_2$ | $P_2O_5$ | $Na_2O$ | $Al_2O_3$ | Other |
|---|---|---|---|---|---|---|---|
| Mass fraction/% | 49.070 | 40.070 | 5.780 | 1.350 | 0.587 | 0.435 | 2.708 |

**Table 5. Test results of heavy metals and radioactivity in phosphogypsum.**

| Test items | | Standard limits | Result | Conclusion |
|---|---|---|---|---|
| Heavy metal | Cu/mg·L$^{-1}$ | ≤100 | 0.157 | Qualified |
| | Zn/mg·L$^{-1}$ | ≤100 | 0.051 | Qualified |
| | Cd/mg·L$^{-1}$ | ≤1 | 0 | Qualified |
| | Pb/mg·L$^{-1}$ | ≤5 | 0 | Qualified |
| | Cr/mg·L$^{-1}$ | ≤15 | 0 | Qualified |
| | As/mg·L$^{-1}$ | ≤5 | 0.0356 | Qualified |
| | Hg/mg·L$^{-1}$ | ≤0.1 | 0.0005 | Qualified |
| Radioactivity | Ra-226/Bq·kg$^{-1}$ | — | 53.94 | — |
| | TH-232/Bq·kg$^{-1}$ | — | 42.13 | — |
| | K-40/Bq·kg$^{-1}$ | — | 52.95 | — |
| | I$_{Ra}$ | ≤1.0 | 0.3 | Qualified |
| | I$_\gamma$ | ≤1.0 | 0.3 | Qualified |

the amount of water will be the remaining water evenly mixed into the material, mixing uniformly, sealed with plastic wrap for one day, so that the mix in the water content is uniform, add the predetermined quality of cement, SCA-2 water stabilizer and the reserved water into the mix, and mix it again. The static method of compression molding: weighing the appropriate amount of mixture poured into the mold, the use of jacks to apply pressure and demolding, the preparation of specimens with a diameter and height of 50 mm. The detailed specimen preparation process is shown in Fig 1.

## Test methods

In this paper, the road performance of CPRC under dry and wet cycling was investigated by using 5% cement as curing agent with the mass ratio of phosphogypsum: red clay = 1:1, plus 5% SCA-2 as water stabilizer, through the unconfined compressive strength test, the CBR test, and the water stability test. The water stabilization mechanism of the mix was analyzed by Scanning Electron Microscope and X-ray diffraction test. The road performance of wet and dry recycled mixes was verified by taking the National Highway G210 Duyun Yang'an-Yingshan Highway Reconstruction and Expansion Project as a test road, which provides a scientific basis for the application of cement-phosphogypsum-red clay on roads.

The test procedure is shown in Fig 2.

The experiments performed in this paper were conducted with permission from the Experimental Center, School of Civil Engineering, Guizhou University, Guizhou, China.

### 4.1 Unconfined compressive strength test

The specimens were prepared in accordance with the method of section 4.2, and after preparation, they were directly placed in the curing box at a temperature of 20±2°C and humidity of ≥95% for 7 days, 14 days and 28 days.

**Table 6. Basic parameters of cement.**

| Item | Index | Item | Index | Item | Index |
|---|---|---|---|---|---|
| 3d $f_{cf}$/MPa | 5.0 | Loss on ignition /% | 1.58 | Initial setting time /min | 302 |
| 28d $f_{cf}$/MPa | 6.7 | Alkali /% | 2.42 | Final setting time /min | 322 |
| 3d $f_{cu}$/MPa | 24.9 | Chloride ion /% | 0.018 | Stability | Qualified |
| 28d $f_{cu}$/MPa | 43.7 | Sulfur trioxide /% | 2.87 | | |

**Table 7. SCA-2 curing agent basic parameters.**

| Morphology of an object | $\rho$ /g.cm-3 | Stickiness /MPa·S | PH value |
|---|---|---|---|
| **Fluids** | **1.06** | **180** | **6.0–8.0** |
| Self-accelerating decomposition temperature /°C | Solubility | Characteristics of the object | Combustion point |
| 60 | Water-soluble | Non-toxic, non-corrosive, non-polluting | Non-combustible |

After the curing of the sample is completed, it is subjected to different cycles of dry and wet cycles, the dry and wet cycle process is described in subsection 3.5. The test is conducted in accordance with T0805-1994 in the Test Methods of Materials Stabilized with Inorganic Binders for Highway Engineering (JTG E51-2009), an unconfined manometer was used to test the compressive strength of the specimens after different numbers of wet and dry cycles. During the test, the specimen is static placed on the moving platform of the manometer, the level of the table is checked and adjusted, the switch is turned on, the table rises at a rate of 1mm/min, and the maximum reading of the percentile meter is observed and recorded at the time of destruction of the specimen. The equation for calculating the unconfined compressive strength of the specimen is as follows:

$$R_c = \frac{P}{A} \tag{1}$$

Note: $R_c$ = Unconfined compressive strength of the specimen for n times of wet and dry cycles (MPa). P = Maximum pressure at destruction of the specimen (N). A = Cross-sectional area of the specimen (mm$^2$).

## Water stability test

The roadbed is directly exposed to the atmosphere and its strength will be significantly reduced by the action of surface and groundwater. Therefore, the roadbed is not only required to have sufficient strength, but also to have sufficient water stability to keep the pavement in a normal stable condition. Therefore, in this paper, we want to study the water stability of mixes, to explore what kind of phenomenon law of its water stability.

Sample preparation was the same as for the unrestricted compressive strength test. According to Luo and Yang [29,30] on water stabilization tests, the specimens were taken out after 6 days of standard curing in the curing chamber to undergo wet and dry cycles, and the wet and dry cycle process is shown in subsection 5.4. At the end of the dry and wet cycle, the specimen was placed in a glass cup containing a certain amount of water, and the water surface did not go over the top of the specimen for about 2~3cm; at regular intervals, photographs were taken to record the changes that occurred in the specimen. Without destroying the specimen, remove the specimen after 24h of soaking and gently dry the surface water to measure its unconfined compressive strength. The specimens with standard curing for 7 days were taken to determine the unconfined compressive strength. The water stability coefficient and appearance test were used as the evaluation criteria for the water stability of the mixture, and the

**Table 8. Mixing ratios.**

| C/% | P/% | T/% | K/% | Optimum moisture content/% | Dosage of water stabiliser/% |
|---|---|---|---|---|---|
| 5 | 47.5 | 47.5 | 96 | 22.9 | 0 |
| | | | | | 5 |

Note: In this paper, C stands for cement, P for phosphogypsum, T for red clay, and K for Compaction degree.

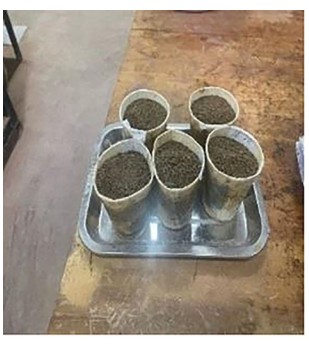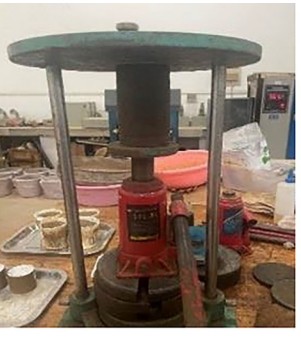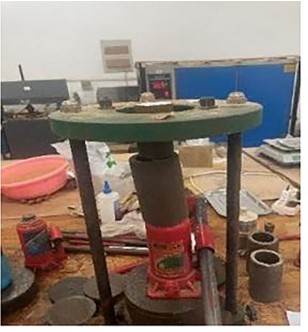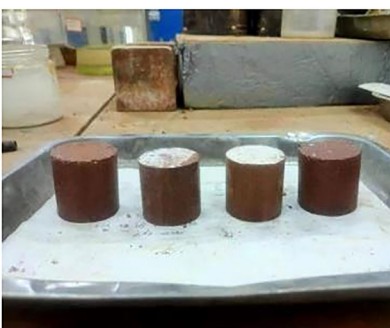

(a) Weighing the mixture    (b) Static compression    (c) Specimen demolding    (d) Sample

molding

**Fig 1. Sample preparation process diagram.**

calculation of water stability coefficient is shown in Eq (2).

$$W = \frac{P_{6d+24h}}{P_{7d}} \qquad (2)$$

Note: W = Coefficient of water stability; $P_{6d+24h}$ = Unconfined compressive strength of standard curing for 6d + soaking in water for 24h; $P_{7d}$ = Unconfined compressive strength at 7d of standard curing.

## California bearing ratio test

According to the "Test Methods of Soils for Highway Engineering" (JTG E40-2007) [31]. The homogeneous mixture was put into the CBR test mould in three times, and each time it was put into the CBR test mould, it was compacted 59 times by using a heavy-duty compactor, so that the specimen plane was consistent with the height of the test bucket. The specimen was taken out after compaction, and placed in the temperature of 20 ± 2°C, humidity ≥ 95% of the conditions of maintenance, maintenance time for 7 days, 14 days, 28 days. Maintenance completed specimen into the bucket, water injection so that the water surface over the top of the test 25mm, immersion for 4 days and nights. After the end of the immersion of the specimen static 15min placed in the CBR strength tester for the test, penetration rod to 1mm per minute speed penetration, penetration distance of 5mm to stop the test, record the instrument readings. The CBR value is calculated as shown in Eq 3.

$$CBR = \frac{P}{7000} \times 100 \qquad (3)$$

Note: CBR = California bearing ratio; P = Pressure (KPa).

## Dry and wet cycle test methods

According to the research of Chen, Tang and other scholars [32,33], the strength decay of the soil body after experiencing first wet and then dry is greater than that after drying and then wetting, so the test adopts the dry-wet cycle process of first wet and then dry. Que, Hu and other scholars concluded [34,35] found that: the roadbed in accordance with the optimal moisture content of the compaction, after a period of time of the natural climate environment, the roadbed soil moisture content will be increased, and in the "equilibrium moisture content" ±

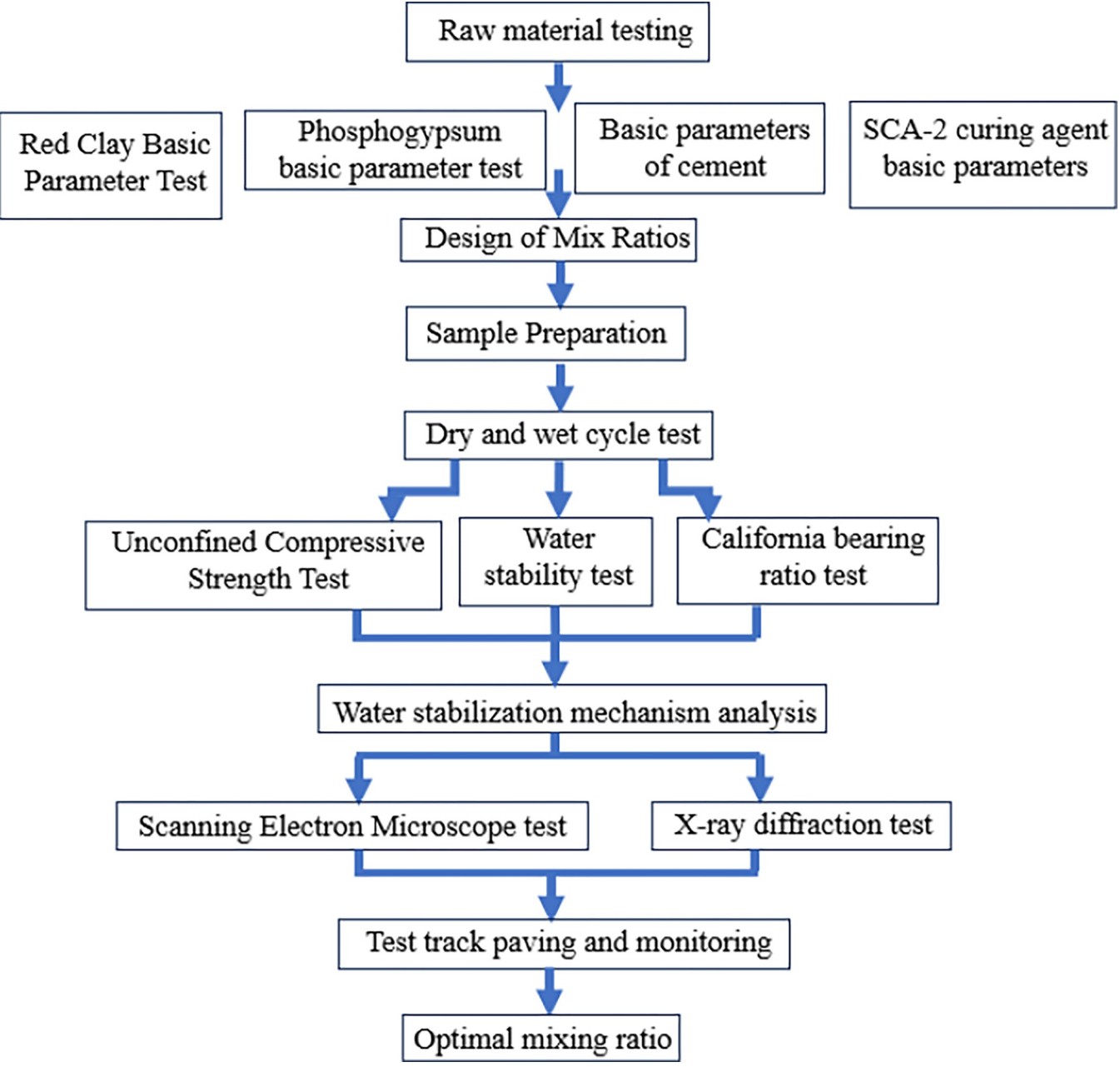

**Fig 2. Flowchart of the testing process.**

5% of the vicinity of the periodic or non-periodic fluctuations, so this paper determines that the wet and dry cycle of the amplitude of 10 percent. Literature [36,37] shows that the strength of soil body tends to stabilise after 5~6 times of wet and dry cycles, so the number of wet and dry cycles is set to 5 times. The wet and dry cycling process is shown in Fig 3.

## Scanning electron microscope and X-ray diffraction tests

The static compaction method was used to prepare the two groups of mixtures into 61.8mm, height of 20mm ring knife specimens, after preparation, they were placed in a constant

temperature and humidity curing box at a temperature of (20±2)°C and humidity of 95% for 7 days, and after 7 days, the specimens were demoulded for the scanning electron microscope test and X-ray diffraction test.

The scanning electron microscope model is HITACHI SU8100 and the accelerating voltage is 3.00kV. The ring knife specimen was broken down into soil pieces close to 2 cm³. The clods were first sanded with coarse sandpaper and then with fine sandpaper to pieces close to 0.5 cm³. Use clear tape to remove the powder particles from the sanding until the surface of the sample is clean. Phosphogypsum-stabilized red clay has poor electrical conductivity and requires a gold spray coating prior to scanning.

The X-ray diffractometer was a Rigaku Smart Lab with a scanning step of 0.02°, a scanning range of 15°~80°, and a scanning rate of 10 (°)/min. The test soil was the soil in the centre of the specimen.

## Test track paving and monitoring

The test road is located at K35+146 of National Highway G210 Duyun Yangan to Yingshan Highway Reconstruction and Expansion Project, adopting the standard of first-class highway, with a design speed of 60km/h, and a paving length of 100m, a width of 3m, and a compaction height of 0.9m. According to the indoor test, the proportion cement: phosphogypsum: red clay = 5: 47.5: 47.5, SCA-2 water stabilizer 5% was selected. The construction process is shown in Fig 4: measurement and placement → foundation treatment → anti-drainage treatment → mixing → transportation paving → leveling → rolling → maintenance.

After the roadbed paving was completed, the location 20m from both ends of the test road and the center position were selected as the settlement monitoring points (e.g., A, B, and C in Fig 5), and the settlement of the roadbed was monitored by using a level meter. The settlement monitoring of the roadbed was measured every 5 days, totaling 7 times.

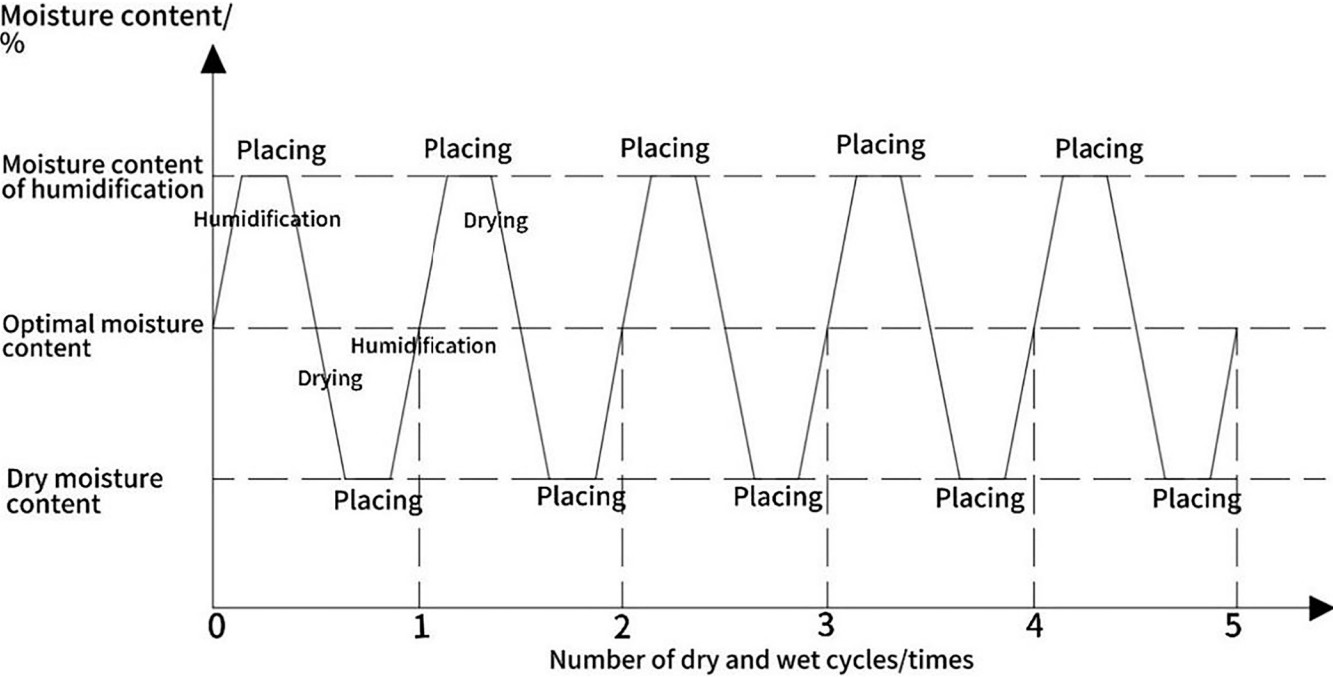

**Fig 3. Schematic diagram of the wet/dry cycle process.**

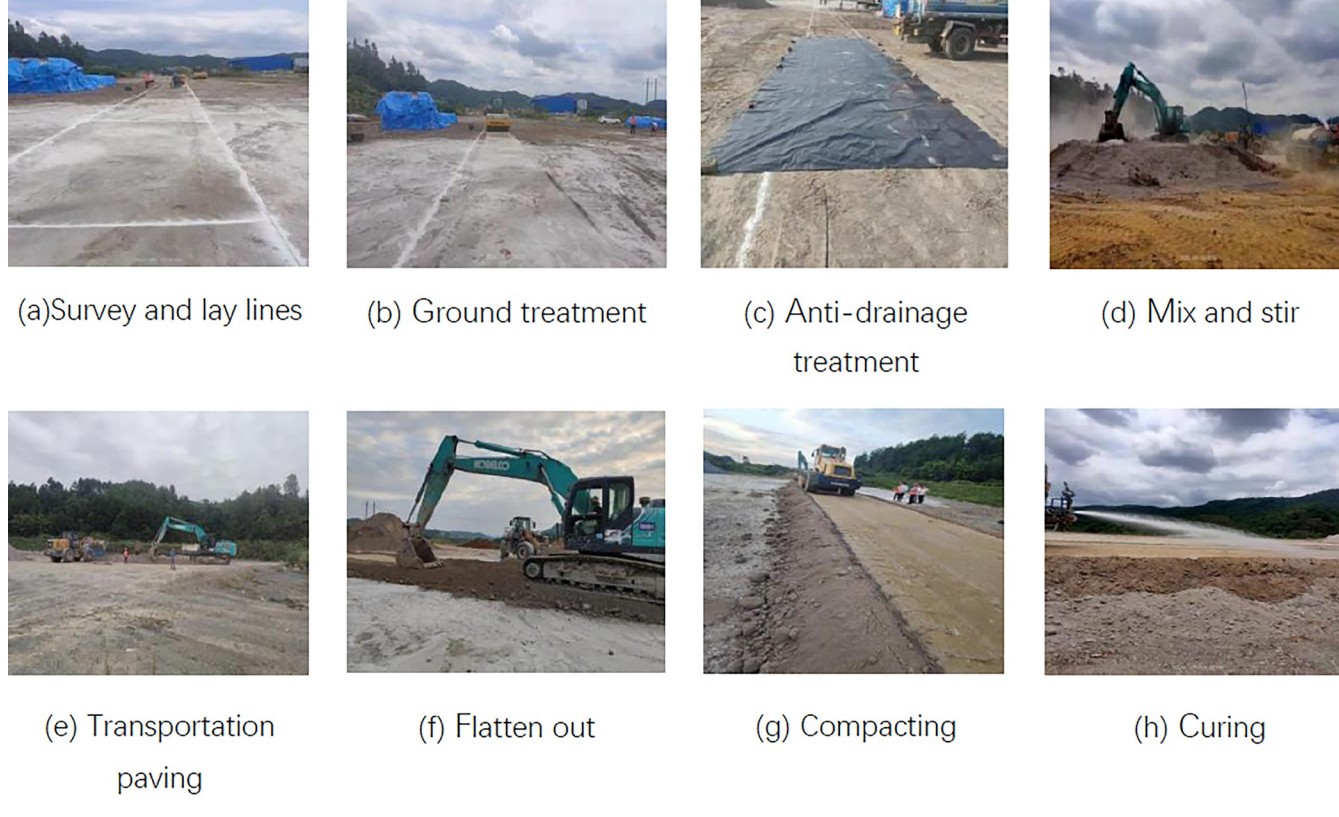

(a)Survey and lay lines (b) Ground treatment (c) Anti-drainage treatment (d) Mix and stir

(e) Transportation paving (f) Flatten out (g) Compacting (h) Curing

**Fig 4. Roadbed construction process.**

## Analysis of test results

### Unconfined compressive strength of mixes

As can be seen from Fig 6, the unconfined compressive strength of the mix decreases with the increase in the number of wet and dry cycles, and the unconfined compressive strength of the mix after the first three wet and dry cycles decreases more, after which the magnitude of the decrease is gentle. Take Fig 6A and Table 9 as examples, the unconfined compressive strength of the mix maintained for 28 days decreased from 2.67MPa at the beginning to 1.02MPa after 3 wet and dry cycles, and the decreased strength was 77.1% of that after 5 wet and dry cycles;

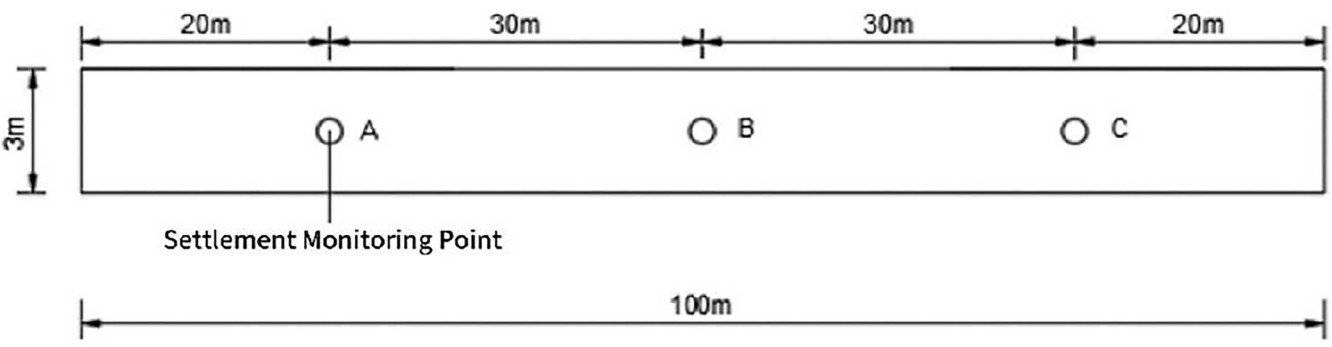

**Fig 5. Layout plan of settlement monitoring points.**

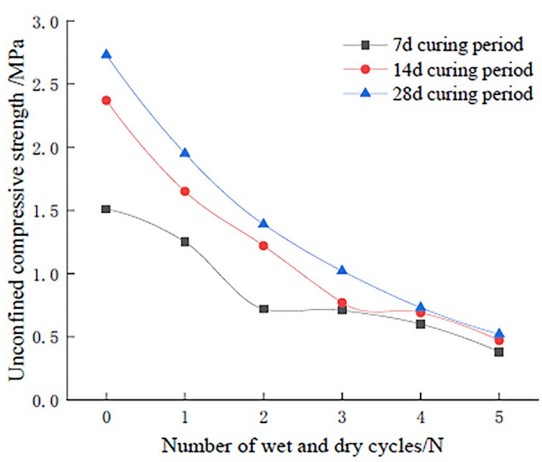
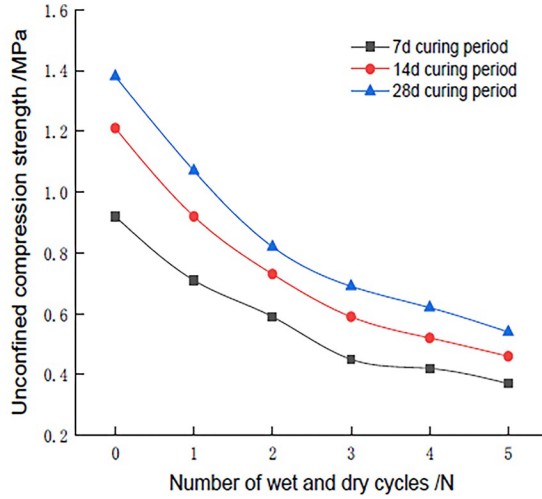

(a)   C: P: T=5: 47.5: 47.5, Unmixed water stabiliser      (b)   C: P: T=5: 47.5: 47.5, Addition of water stabiliser

**Fig 6. Relationship between unconfined compressive strength and number of wet and dry cycles.**

whereas, it decreased to 0.49MPa after 3~5 wet and dry cycles, which was 22.9% of that after 5 wet and dry cycles, and the decreased strength was obviously lower than that of the first 3 wet and dry cycles. Mixed material in the first three wet and dry cycle process, due to wet expansion and contraction of a large number of cracks, so that the soil body separated into pieces, most of the soil body of the cementitious material is damaged, cohesion decreases, which leads to a sharp decline in the unconfined compressive strength [38]. Three wet and dry cycle, most of the soil body of the cementitious material has been damaged, the remaining small portion of the cementitious material of the soil body strength is not much, so with the increase in the number of wet and dry cycle, the unconfined compressive strength attenuation gradually decreases. Fig 6B has the same conclusions as Fig 6A and is therefore not illustrated further. It was also verified in the study of Ren [39] and Cui [40] that the unconfined compressive strength of modified red clay decreased gradually with the increase of the number of wet and dry cycles, and leveled off after the third cycle. This paper further confirmed that the strength properties of the highly doped phosphogypsum stabilized materials meet the strength requirements, which reduces the damage of the phosphogypsum pollution of the environment to a certain degree, and solves the problem of the low-doped phosphogypsum road base layer material. It is difficult to meet the problem of large-scale consumption of phosphogypsum demand, and it is feasible for subsequent research.

**Table 9. C: P: T = 5: 47.5: 47.5, numerical value of unconfined compressive strength without water stabilizers/ MPa.**

| Number of wet and dry cycles/N | 7d curing period | 14d curing period | 28d curing period |
| --- | --- | --- | --- |
| 0 dry wet cycles | 1.51 | 2.33 | 2.67 |
| 1 dry wet cycle | 1.22 | 1.61 | 1.88 |
| 2 dry wet cycles | 0.69 | 1.18 | 1.48 |
| 3 dry wet cycles | 0.66 | 0.73 | 1.02 |
| 4 dry wet cycles | 0.58 | 0.62 | 0.87 |
| 5 dry wet cycles | 0.31 | 0.41 | 0.53 |

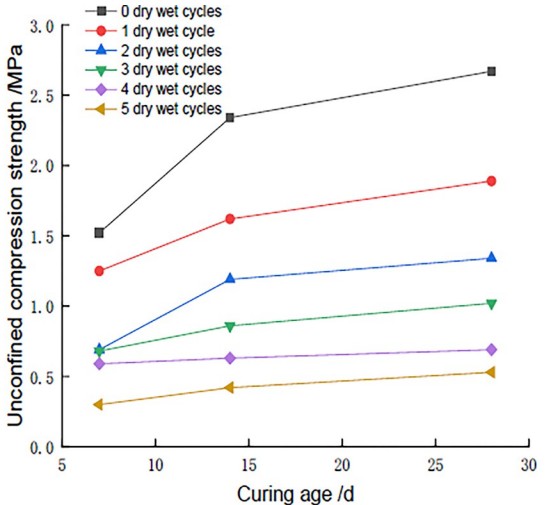
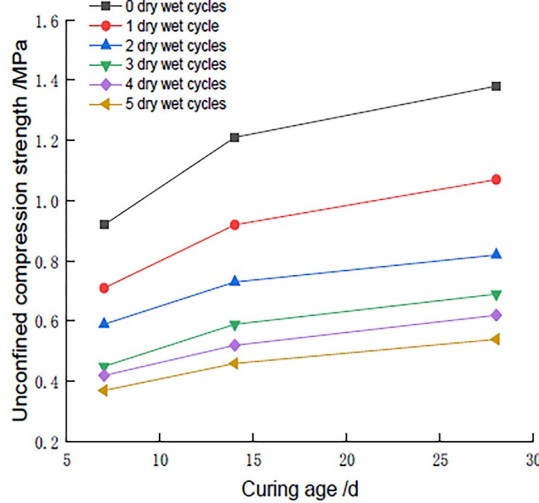

(a)   C: P: T=5: 47.5: 47.5， Unmixed water stabiliser

(b)   C: P: T=5: 47.5: 47.5， Addition of water stabiliser

**Fig 7. Relationship between unconfined compressive strength and age.**

As can be seen from Fig 7 and Tables 9 and 10, the unconfined compressive strength of the mixes all increased with age, and the unconfined compressive strength at the age of 14 days reached more than 60% of the strength at the age of 28 days. For example according to Tables 9 and 10, the two groups of mixes mixed with and without water stabilizer were maintained for 14 days, and the unconfined compressive strength was 0.73MPa and 1.18MPa after 2 dry and wet cycles, and the strengths were 0.81MPa and 1.48MPa after 2 dry and wet cycles for 28 days, and the unconfined compressive strengths at the age of 14 days reached 90.1% and 79.7%, respectively, at the age of 28 days. According to Luo [29], it also shows that the compressive strength of the mix increases with age. With the increase of age, the hydration reaction of cement continues, and the generated gel substances such as hydrated calcium silicate and the mix undergoes cementation, the structural stability is enhanced, and the compressive strength is increased.

As can be seen from Fig 8, the unconfined compressive strength of the mix without water stabiliser is significantly higher than that of the mix with 5% SCA-2 water stabiliser in the first three cycles, and gradually tends to be closer thereafter. The hydration reaction between cement and phosphogypsum inside the mix produces cementitious substances and ettringite which are responsible for the increase in compressive strength of the mix. According to Table 12 in Section 6.2, a table of compositional analysis of mixes maintained for 7 days, it can be seen that the relative content of gypsum in the mixture decreased from 86.5% to 59.3% after

**Table 10. C: P: T = 5: 47.5: 47.5, unconfined compressive strength values of water-doped stabilizers/ MPa.**

| Number of wet and dry cycles/N | 7d curing period | 14d curing period | 28d curing period |
|---|---|---|---|
| 0 dry wet cycles | 0.92 | 1.21 | 1.35 |
| 1 dry wet cycle | 0.71 | 0.92 | 1.07 |
| 2 dry wet cycles | 0.61 | 0.73 | 0.81 |
| 3 dry wet cycles | 0.45 | 0.61 | 0.69 |
| 4 dry wet cycles | 0.42 | 0.53 | 0.62 |
| 5 dry wet cycles | 0.38 | 0.47 | 0.54 |

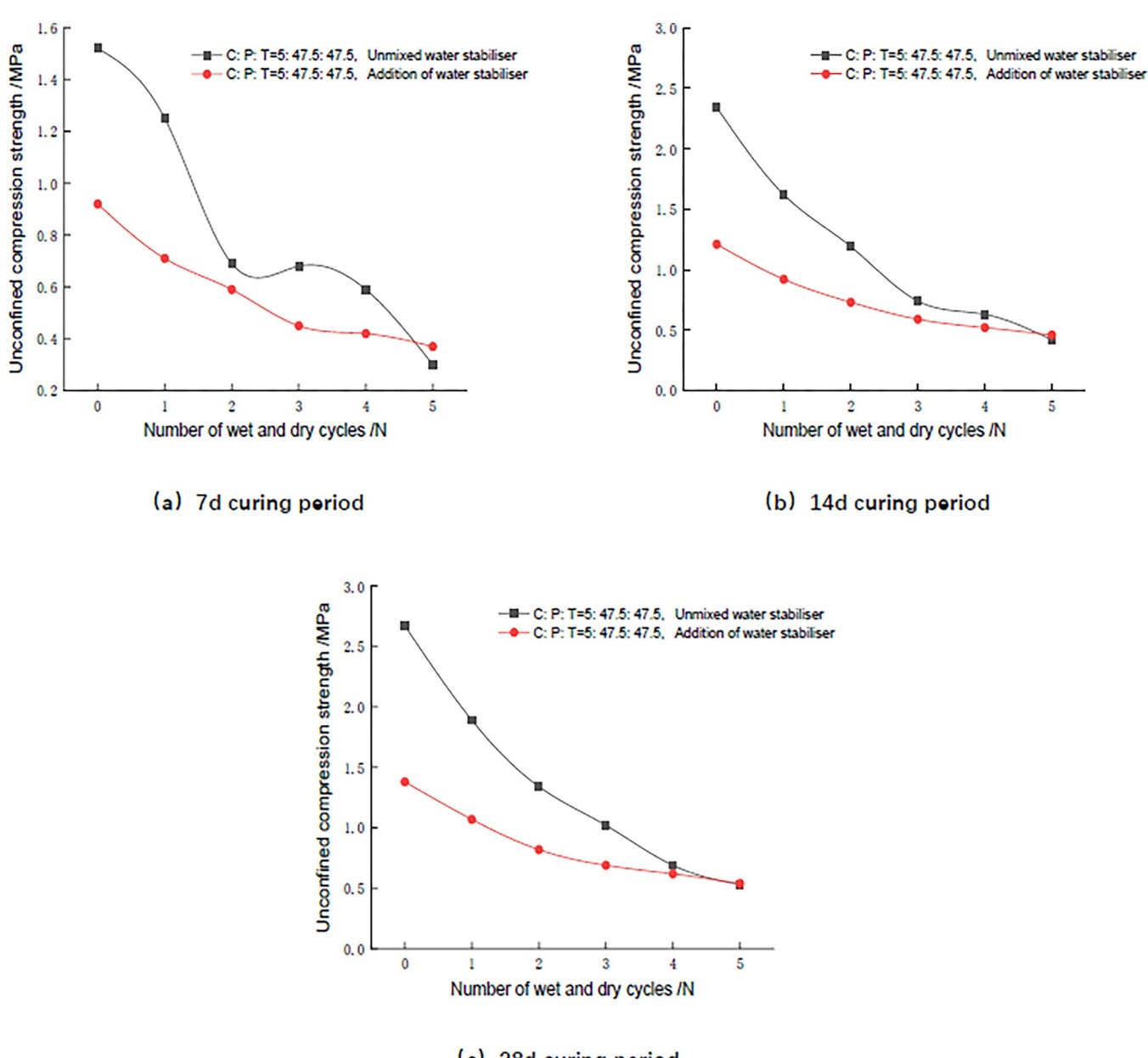

**Fig 8. Relationship between the unconfined compressive strength of specimens and the number of wet and dry cycles at different curing ages.**

the addition of SCA-2 water stabiliser, and the decrease in gypsum content led to a decrease in the production of ettringite, which resulted in a lower strength of the mixture with SCA-2 water stabiliser in the initial state compared with that of the mixture without water stabiliser. In Section 5.2, comparing the quantitative analysis of mix composition in Table 12, it is also clear that the amount of ettringite in the SCA-2 water stabilizer mix is lower than the amount in the unadulterated mix. After three wet and dry cycles, the cementitious substances and ettringite in the mixes are damaged, and the strengths are drastically decayed. The content of ettringite in the initial state has a weak effect on the strength of the mixes at this time, so with the increase of the number of wet and dry cycles, the unconfined compressive strengths of the two groups of mixes are closer and closer to each other.

Table 12. Quantitative analysis table of mix composition.

| Mineral phase content/ wt% | C: P: T = 5: 47.5: 47.5, Unmixed water stabiliser | C: P: T = 5: 47.5: 47.5, Addition of water stabiliser |
|---|---|---|
| $SiO_2$ | 4.9 | 11.6 |
| $CaCO_3$ | - | 1.1 |
| $CaSO_4 \cdot 2H_2O$ | 86.5 | 59.3 |
| $FeO \cdot OH$ | 0.8 | 4.0 |
| $(K, Na)Al_2(Si, Al)_4O_{10}(OH)_2$ | 3.5 | 10.4 |
| $Al_2(OH)_4Si_2O_5$ | 2.5 | 12.0 |
| $Ca_6Al_2(SO_4)3(OH)_{12} \cdot 26H_2O$ | 1.8 | 1.6 |

## Water stability properties of mixes

According to Fig 9, it can be found that when SCA-2 water stabiliser was not mixed, the bottom periphery of the specimen appeared to be peeled off after being immersed in water for 30 s, and the bottom began to swell gradually. When immersed for 3 min, the middle part of the specimen appeared to fall obviously, the number of cracks at the bottom increased a lot, and the specimen was destroyed, indicating that the specimen without water stabiliser did not have water stability properties. As shown in Fig 10, after mixing SCA-2 water stabiliser, there is no cracks, swelling, falling and other phenomena after the specimen is immersed in water for 30 s. When immersed in water for 24 h, a large number of small bubbles appear on the surface of the specimen, and when the specimen is gently pressed by a finger, it can be found that the specimen has a certain degree of strength. It can be seen that after adding SCA-2 water stabilizer, the water stability performance of the sample is significantly improved. From Section 4.5, this is because the active components in SCA-2 water stabilizer react with soil particles, transforming the layered and colloidal structures in the soil into mineral lattice structures. Through chemical reactions, the physical structure and chemical properties of soil particles and organic matter are changed, reducing the hydrophilicity of the soil itself, and the arrangement of soil particles is more dense, the repulsion between particles decreases.

As can be seen from Fig 11 and Table 11, after 0~2 times of wet and dry cycles, the SCA-2 water stabiliser doped specimens remained intact after immersion in water for 24h without

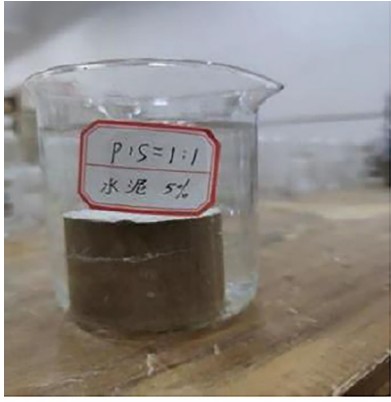

(a) Soak in water for 0 seconds

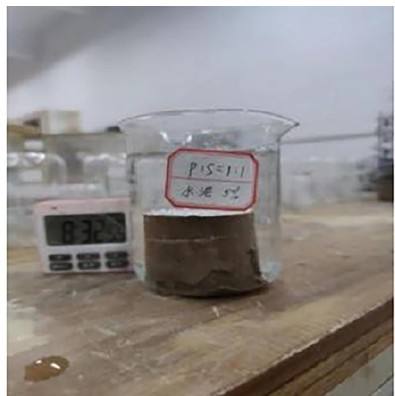

(b) Soak in water for 30 seconds

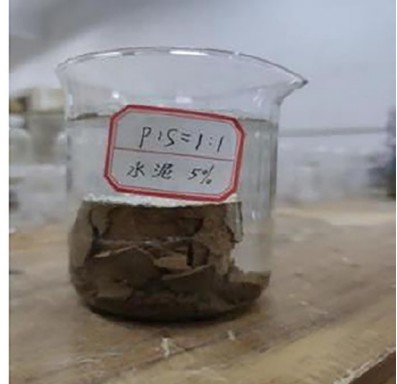

(c) Soak in water for 3 minutes

**Fig 9. Undoped SCA-2 water stabilizer specimen standard maintenance for 6 days.**

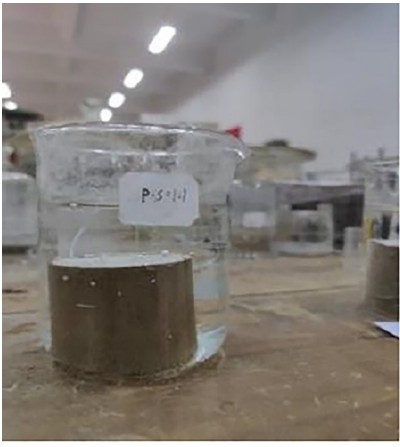
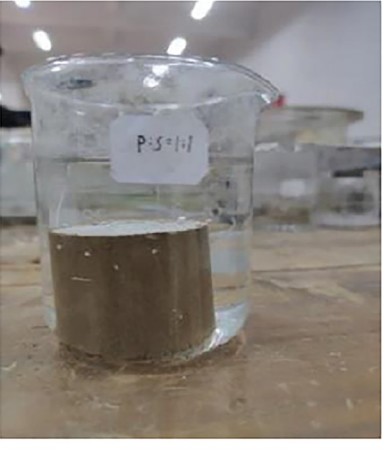
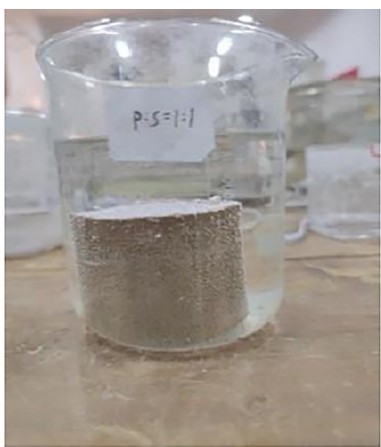

(a) Soak in water for 0 seconds　　(b) Soak in water for 30 seconds　　(c) Soak in water for 24 hours

**Fig 10. SCA-2 water stabilizer doped specimen standard maintenance for 6 days.**

obvious damage, and the coefficient of water stability decreased from 84.52% after 0 times of wet and dry cycles to 38.88% after 2 times of wet and dry cycles, and a large number of bubbles appeared on the surface of the specimens after 2 times of wet and dry cycles; The specimen was immersed in water for 30 min after 3 wet and dry cycles, the bottom of the specimen began to break, and the coefficient of water stability was reduced to 0%; after 4 wet and dry cycles the specimen was only immersed in water for 1 min and then there was a large area of destruction. It can be seen that with the increase in the number of wet and dry cycles, the water stability of the specimen decreases, and the specimen after 3 wet and dry cycles begins to have no water stability. According to Peng et al. [41] showed that the coefficient of water

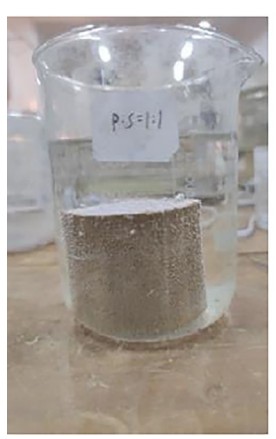
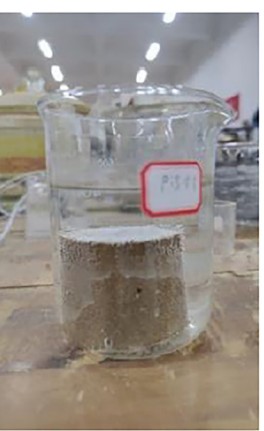
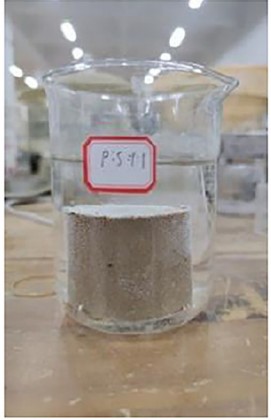
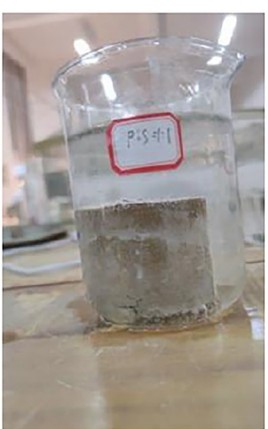
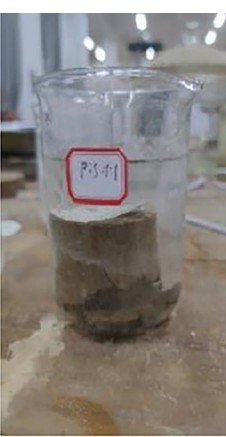

(a) Soaking in water for 24h after 0 wet/dry cycles　　(b) Soaking in water for 24h after 1 wet/dry cycle　　(c) Soaking in water for 24h after 2 wet/dry cycle　　(d) Soak in water for 30min after 3 wet/dry cycles.　　(e) Soak in water for 1min after 4 wet/dry cycles.

**Fig 11. SCA-2 water stabilizer doped samples under different wet and dry cycles.**

**Table 11. Table of water stability coefficients of cement phosphogypsum stabilized red clay mixes.**

| mixing ratio | K/% | Number of dry and wet cycles | coefficient of water stability/% |
|---|---|---|---|
| C: P: T = 5:47.5:47.5, Unmixed water stabiliser | 96 | 0 | 0 |
| | | 1 | 0 |
| | | 2 | 0 |
| | | 3 | 0 |
| | | 4 | 0 |
| | | 5 | 0 |
| C: P: T = 5: 47.5: 47.5, Addition of 5% SCA-2 water stabiliser | 96 | 0 | 84.52 |
| | | 1 | 55.55 |
| | | 2 | 38.88 |
| | | 3 | 0 |
| | | 4 | 0 |
| | | 5 | 0 |

stability of the mix decreases with the increase of the number of wet and dry cycles, due to the wet expansion and dry contraction will produce cracks, the collodion material in the specimen is damaged, and thus the strength decreases leading to the decrease of water stability.

In order to investigate the reason why the water stability of the specimens became stronger after the SCA-2 water stabilizer was doped, the scanning electron microscope test (magnification of 10k) and X-ray diffraction test were subsequently carried out, and the results are shown in Fig 12 and Table 12.

As shown in Fig 12A, an exposed plate-like substance—phosphogypsum can be seen inside the mixture without the addition of water stabilizer. At the same time, it is also observed that some irregular cementitious substances cover the phosphogypsum, causing the internal surface of the mixture to be uneven, irregular in shape, and loosely arranged in internal structure. From Fig 12B, it can be seen that after the addition of SCA-2 water stabilizer, the internal morphological characteristics of the mixture have undergone significant changes, with reduced surface undulation and becoming smoother. The amount of cementitious material increases and adheres to the surface of the mixture, tightly wrapping it. This is because the active ingredient in SCA-2 water stabilizer reacts with soil particles, transforming the layer structure and

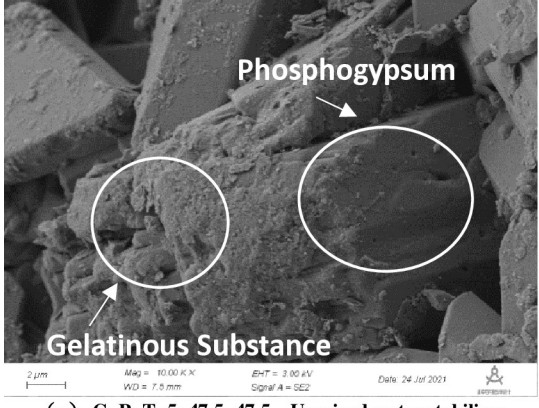
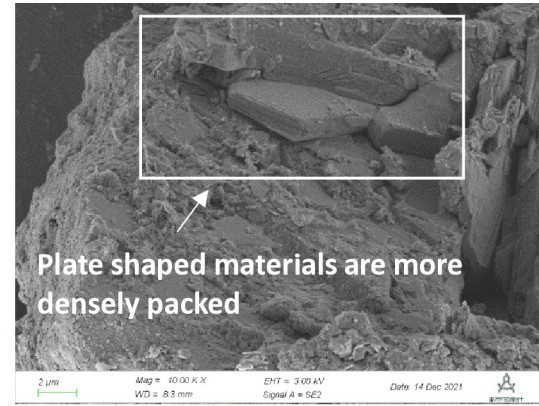

（a）C: P: T=5: 47.5: 47.5，Unmixed water stabiliser        （b）C: P: T=5: 47.5: 47.5，Addition of water stabiliser

**Fig 12. Electron microscope scanning image of the mix.** a) C: P: T = 5: 47.5: 47.5,Unmixed water stabiliser, b) C: P: T = 5: 47.5: 47.5, Addition of water stabiliser.

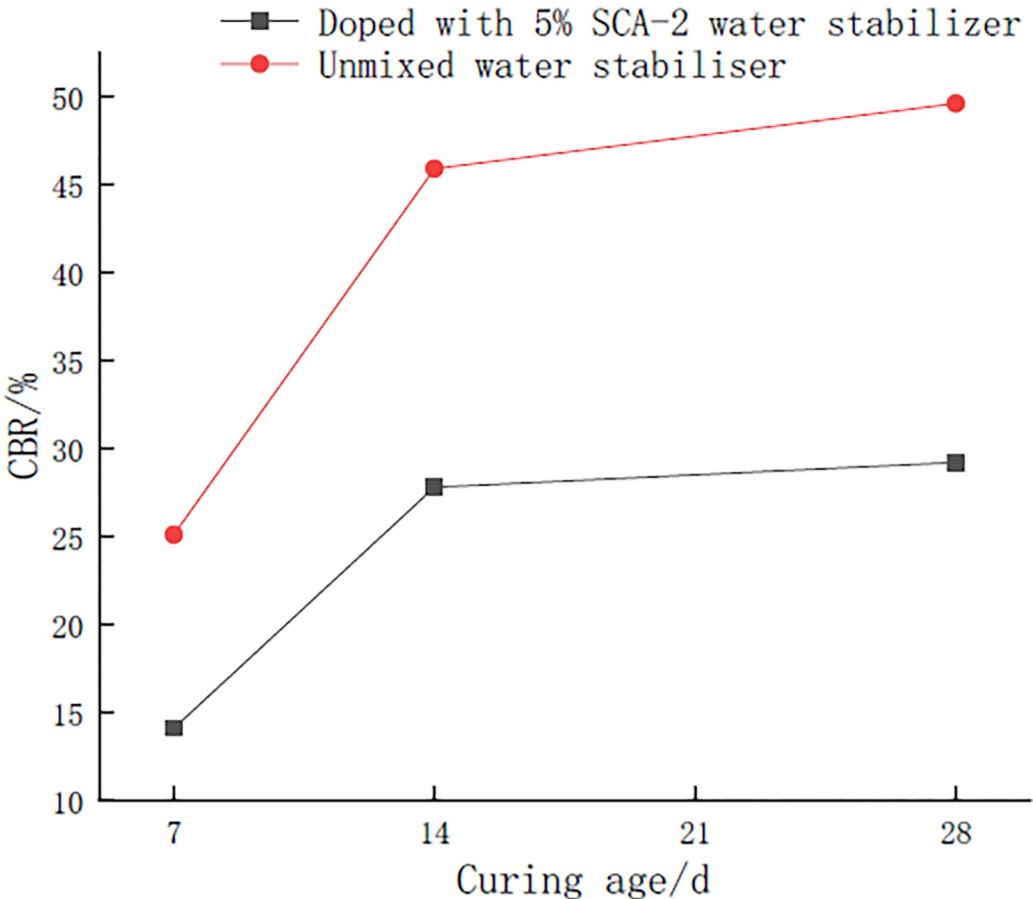

**Fig 13. CBR values of mixes at different maintenance ages.**

gelatinous structure of the soil into mineral lattice structure, and the soil particles and organic matter are chemically reacted to change their physical structure and chemical properties, reducing the hydrophilicity of the soil itself, making the arrangement of soil particles more compact and reducing the repulsive force between particles.

As the phosphogypsum cement stabilized red clay in the process of dry and wet cycle, the hydration reaction of cement occurs, resulting in C-S-H gel [42], the chemical reaction formula as in Eq (4), and phosphogypsum main component calcium sulfate dihydrate ($CaSO_4 \cdot 2H_2O$), reacts with the hydration products of cement to produce ettringite, the chemical reaction formula as in Eqs (5) and (6).

$$3CaO \cdot SiO_2 + nH_2O \rightarrow xCaO \cdot SiO_2 \cdot (n - 3 + x)H_2O + (3 - x)Ca(OH)_2 \tag{4}$$

$$3Ca(OH)_2 + Al_2O_3 + (n - 3)H_2O \rightarrow 3CaAl_2O_3 \cdot nH_2O \tag{5}$$

$$3CaO \cdot Al_2O_3 + 3(CaSO_4 \cdot 2H_2O) + 2Ca(OH)_2 + 24H_2O \rightarrow 3\,CaO \cdot Al_2O_3 \cdot 3CaSO_4 \cdot 32H_2O \tag{6}$$

The newly generated material is very dense inside the mixture, blocking the loose permeable channels, reducing the number and volume of pores, forming a dense system, thereby improving water stability. And comparing Fig 12A and 12B, it can be found that the mixture is irregularly arranged although it is a large number of plate-like arrangement when no

**Table 13. Mix CBR value/%.**

| Compaction degree | 7d curing period | 14d curing period | 28d curing period | Note |
|---|---|---|---|---|
| 96% compaction degree | 25.1 | 45.9 | 49.6 | Unmixed water stabilizer |
| | 14.1 | 27.8 | 29.2 | Doped with 5% SCA-2 water stabilizer |

admixture is added, so that although there is a certain degree of water stability, but there are more pores, so the water stability is not strong. After the addition of admixtures, the plate-like stacking is more regular, which will also improve the water stability. From the composition analysis in Table 12, it can be seen that after mixing the curing agent, quartz, mica and kaolinite and other poor hydrophilic substances in the mixture have different magnitudes of increase, which to a certain extent weakened the mix in the soaking process of the water absorption capacity, thus reducing the expansion and deformation due to the absorption of water caused by the mix to enhance the stability of the internal structure of the mix, the stability of the water has been improved.

**CBR values of mixes.** From Fig 13 and Table 13, it can be seen that the CBR value of the mixture is much larger than the requirement of 8% CBR value specified for roadbed filler on highway and primary roadbed as stipulated in Highway Roadbed Design Specification (JTG D30-2015), no matter whether it is doped with water stabilizing grades or not doped with water stabilizers. According to Table 13 and Fig 13, it can be seen that the CBR values of the mixes first increase and then gradually level off as the age of maintenance develops, and the 14-day CBR values of the mixes are about 90% or more of those of the 28-day values, mixed with and without the SCA-2 water stabilizer. After the addition of SCA-2 water stabilizer, the CBR value of the mix was reduced and the mechanism was consistent with the unconfined compressive strength. According to the research results of Chen et al. [43] on the CBR properties of cement-phosphogypsum-stabilized red clay, the accuracy of the present conclusions is more confirmed. For cement-phosphogypsum-red clay used as fill for road beds on highway subgrade for Class I roads, not only the cost of cement is saved, but phosphogypsum is also highly used and the ecological environment is better protected.

## Test road monitoring

According to Table 14 and Fig 14A, it can be seen that the settlement of roadbed increases gradually with the continuation of time and tends to be stabilized, and the final settlements of points A, B and C are 7mm, 9mm and 10mm respectively. It is much smaller than the requirement of "Highway Roadbed Design Specification" (JTGD30-2015) that the post-work settlement of general roadbed should not exceed 30cm. As can be seen from the examination of the pavement appearance in Fig 14B, the surface of the test road is smooth and dense, and there is no scouring, peeling, cracking or obvious settlement phenomenon.

According to the research of Gaoguo Ke [44], it can be seen that in order to verify the road performance of phosphogypsum, a test road study can be carried out to explore the feasibility of the mixture in the actual project and to detect the road performance, and this research

**Table 14. Mixed embankment settlement monitoring results.**

| Measurement time (Date) | | 2021.8.3 | 2021.8.5 | 2021.8.8 | 2021.8.12 | 2021.8.18 | 2021.8.25 | 2021.9.7 | 2021.9.15 |
|---|---|---|---|---|---|---|---|---|---|
| Measurement point A | Settlement (mm) | 3 | 3 | 1 | 0 | 0 | 0 | 0 | 0 |
| Measurement point B | | 3 | 2 | 1 | 1 | 1 | 1 | 0 | 0 |
| Measurement point C | | 4 | 3 | 2 | 1 | 0 | 0 | 0 | 0 |

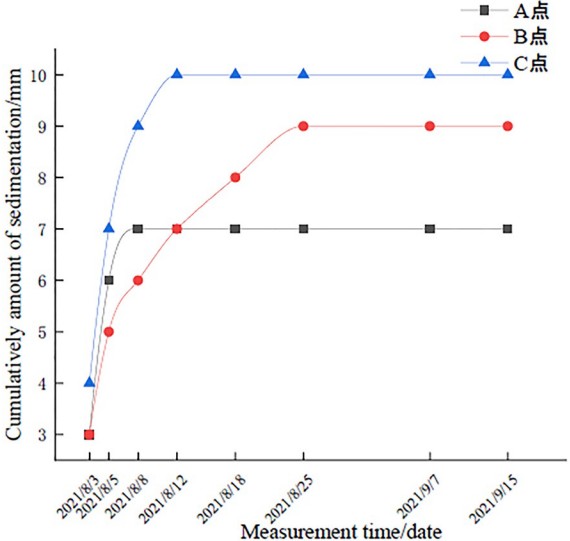

(a) Plot of settlement versus number of measurements

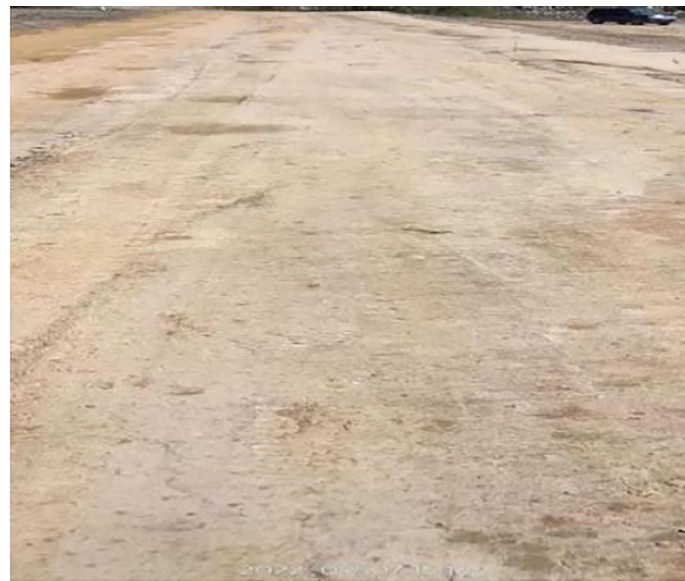

(b) Pavement appearance inspection

**Fig 14. Test road monitoring results.**

method only measured the flatness and strength of the test road to detect the road performance. According to Liu Chao's research [12] on cement phosphogypsum stabilized crushed stone base layer, it was detected through the test road to meet the basic design requirements of high grade highway. In this paper, through the settlement monitoring and appearance inspection of the test road, and compared with the previous results, combined with the requirements of the "Highway Roadbed Design Specification" (JTGD30-2015), this method further examines whether the road performance of phosphogypsum-stabilized red clay meets the requirements.

## Conclusion

In this paper, the road performance of CPRC under dry and wet cycling was investigated by using 5% cement as curing agent with the mass ratio of phosphogypsum: red clay = 1:1, plus 5% SCA-2 as water stabilizer, through the unconfined compressive strength test, the California bearing ratio (CBR) test, and the water stability test. The water stabilization mechanism of the mixture was analyzed by Scanning Electron Microscope and X-ray diffraction tests. The road performance of dry and wet cycle mix was verified with the National Highway G210 Duyun Yangan to Yingshan Highway Reconstruction and Expansion Project as a test road to provide a scientific basis for the application of cement-phosphogypsum-red clay on roads. The following conclusions were reached:

1. The CPRC unconfined compressive strength decreases with the number of wet and dry cycles, with a large decay in the first three and leveling off thereafter. The unconfined compressive strength increased with age, reaching more than 60% of the 28-day unconfined compressive strength at 14 days of age. The CBR value fully meets the requirements of roadbed fill on highway and primary roadbed as stipulated in the Design Code for Highway Roadbeds (JTG D30-2015).

2. SCA-2 water stabilizer mixed with CPRC reduces the strength of the mixture, but obviously improves the water stability performance of the mixture. SCA-2 water stabilizer changes the internal structure of the mixture, the structure is more dense, and the gelling material increases and adheres to the surface of the mixture, which tightly wraps the mixture and prevents water from seeping into the interior, and the hydrophilicity of the quartz, white mica, and kaolinite and other poorly hydrophilic substances have increased by varying degrees. The hydrophilic substances such as quartz, dolomite and kaolinite are all increased in different magnitudes.

3. Through the observation of settlement and appearance inspection of the test road, the road-bed is smooth and dense, and there is no scouring, detachment and cracking. The settlement of the roadbed is far less than the requirement of "Highway Roadbed Design Specification" (JTGD30-2015) that the settlement of general roadbed is not more than 30cm after work.

4. Through the CPRC road performance test and test road monitoring, CPRC (cement 5%, phosphogypsum 47.5%, red clay 47.5%, SCA-2 water stabilizer 5%) can be used as the filler for the roadbed on the roadbed of the highway and the first-class highway.

Future Analysis:

The main ways of comprehensive utilization of phosphogypsum resources include mine filling and ecological restoration materials, road construction materials, soil conditioning agents, cement retarders, gypsum-based building materials, sulphuric acid production, chemical fillers and so on. Since 2020, desulfurization of phosphogypsum comprehensive utilization of the amount of a declining trend, comprehensive utilization of less than 50%, in a variety of ways to use, phosphogypsum used for road construction materials less than 10%, and the demand for raw materials for the huge demand for road projects, obviously the amount is too small. Restricted by the temporary absence of successful experience in large-scale application, as well as environmental protection and other administrative licenses and related policy support and other factors, phosphogypsum as a road construction material has not yet opened up the market, not to achieve engineering, large-scale application. The reasons are as follows:

1. Lack of a clear administrative license basis, there is currently no clear environmental protection and other relevant administrative licenses as a basis.

2. Technically feasible, but with limited transportation distance, economy to be further cost reduction, and small product coverage radius.

3. Phosphogypsum in the road project in the application of low admixture, the accumulation of the problem is still not effectively solved, an important reason for the lack of technical support capacity, there is no formation of phosphogypsum in the road and other aspects of the systematic system of standards.

Future Research Directions: (1) Increase the basic and forward-looking technology research and development funding for the comprehensive utilization of phosphogypsum, and there is no systematic standard system covering phosphogypsum emissions, comprehensive utilization technology, comprehensive utilization products and other aspects. (2) Clarify the basis for administrative licenses. Relevant departments to clarify the environmental protection and other relevant administrative licenses as the basis for relevant policies and safeguards to guide, support and support. (3) Through the research of this paper, the phosphogypsum mixing amount reaches 47.5%, phosphogypsum is adopted as the main material of highway construction, which changes the problem of low utilization rate in the past (generally not more than

15%), not only solves the problem of phosphogypsum dumping, but also improves the engineering characteristics of the red clay, and it is recommended to popularize the application.

## Supporting information

**S1 Dataset.**
(DOCX)

## Author Contributions

**Conceptualization:** Yunke Liu, Kaisheng Chen.

**Data curation:** Yunke Liu, Hao Jian, Zeyu Liu.

**Formal analysis:** Yunke Liu, Kaisheng Chen, Hao Jian.

**Writing – original draft:** Yunke Liu.

**Writing – review & editing:** Yunke Liu, Kaisheng Chen, Hao Jian.

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
