## [Decision Letter · Decision Letter 0]

8 Oct 2024

PONE-D-24-26244Research on road performance and crack extension law of cement-phosphogypsum-red clay under dry and wet cyclePLOS ONE

Dear Dr. Chen,

Thank you for submitting your manuscript to PLOS ONE. After careful consideration, we feel that it has merit but does not fully meet PLOS ONE’s publication criteria as it currently stands. Therefore, we invite you to submit a revised version of the manuscript that addresses the points raised during the review process.

We look forward to receiving your revised manuscript.

Kind regards,

Nezha Mejjad, PhD

Academic Editor

PLOS ONE

Journal requirements: When submitting your revision, we need you to address these additional requirements. 1. Please ensure that your manuscript meets PLOS ONE's style requirements, including those for file naming. The PLOS ONE style templates can be found at https://journals.plos.org/plosone/s/file?id=wjVg/PLOSOne_formatting_sample_main_body.pdf and https://journals.plos.org/plosone/s/file?id=ba62/PLOSOne_formatting_sample_title_authors_affiliations.pdf 2. In your Methods section, please provide additional information regarding the permits you obtained for the work. Please ensure you have included the full name of the authority that approved the field site access and, if no permits were required, a brief statement explaining why. 3. We suggest you thoroughly copyedit your manuscript for language usage, spelling, and grammar. If you do not know anyone who can help you do this, you may wish to consider employing a professional scientific editing service.  The American Journal Experts (AJE) (https://www.aje.com/) is one such service that has extensive experience helping authors meet PLOS guidelines and can provide language editing, translation, manuscript formatting, and figure formatting to ensure your manuscript meets our submission guidelines. Please note that having the manuscript copyedited by AJE or any other editing services does not guarantee selection for peer review or acceptance for publication.  Upon resubmission, please provide the following: The name of the colleague or the details of the professional service that edited your manuscript A copy of your manuscript showing your changes by either highlighting them or using track changes (uploaded as a *supporting information* file) A clean copy of the edited manuscript (uploaded as the new *manuscript* file)”. 4. Thank you for stating the following financial disclosure:  [Guizhou Provincial Science and Technology Program (Qiankehe Basic-ZK [2023] Key 016); Guiyang Municipal Science and Technology Program Project (Zhuke Contract [2024]-1-12).].  Please state what role the funders took in the study.  If the funders had no role, please state: ""The funders had no role in study design, data collection and analysis, decision to publish, or preparation of the manuscript."" If this statement is not correct you must amend it as needed. Please include this amended Role of Funder statement in your cover letter; we will change the online submission form on your behalf. 5. Thank you for stating the following in the Acknowledgments Section of your manuscript: [This work was supported by Guizhou Provincial Science and Technology Program (Qiankehe Basic-ZK [2023] Key 016); Guiyang Municipal Science and Technology Program Project (Zhuke Contract [2024]-1-12).]We note that you have provided funding information that is not currently declared in your Funding Statement. However, funding information should not appear in the Acknowledgments section or other areas of your manuscript. We will only publish funding information present in the Funding Statement section of the online submission form. Please remove any funding-related text from the manuscript and let us know how you would like to update your Funding Statement. Currently, your Funding Statement reads as follows:   [Guizhou Provincial Science and Technology Program (Qiankehe Basic-ZK [2023] Key 016); Guiyang Municipal Science and Technology Program Project (Zhuke Contract [2024]-1-12).].   Please include your amended statements within your cover letter; we will change the online submission form on your behalf. 6. We note that you have indicated that there are restrictions to data sharing for this study. PLOS only allows data to be available upon request if there are legal or ethical restrictions on sharing data publicly. For more information on unacceptable data access restrictions, please see http://journals.plos.org/plosone/s/data-availability#loc-unacceptable-data-access-restrictions.  Before we proceed with your manuscript, please address the following prompts: a) If there are ethical or legal restrictions on sharing a de-identified data set, please explain them in detail (e.g., data contain potentially identifying or sensitive patient information, data are owned by a third-party organization, etc.) and who has imposed them (e.g., a Research Ethics Committee or Institutional Review Board, etc.). Please also provide contact information for a data access committee, ethics committee, or other institutional body to which data requests may be sent. b) If there are no restrictions, please upload the minimal anonymized data set necessary to replicate your study findings to a stable, public repository and provide us with the relevant URLs, DOIs, or accession numbers. For a list of recommended repositories, please seehttps://journals.plos.org/plosone/s/recommended-repositories. You also have the option of uploading the data as Supporting Information files, but we would recommend depositing data directly to a data repository if possible. We will update your Data Availability statement on your behalf to reflect the information you provide. 7. We note that your Data Availability Statement is currently as follows: [All relevant data are within the manuscript and its Supporting Information files.] Please confirm at this time whether or not your submission contains all raw data required to replicate the results of your study. Authors must share the “minimal data set” for their submission. PLOS defines the minimal data set to consist of the data required to replicate all study findings reported in the article, as well as related metadata and methods (https://journals.plos.org/plosone/s/data-availability#loc-minimal-data-set-definition). For example, authors should submit the following data: - The values behind the means, standard deviations and other measures reported;- The values used to build graphs;- The points extracted from images for analysis. Authors do not need to submit their entire data set if only a portion of the data was used in the reported study. If your submission does not contain these data, please either upload them as Supporting Information files or deposit them to a stable, public repository and provide us with the relevant URLs, DOIs, or accession numbers. For a list of recommended repositories, please see https://journals.plos.org/plosone/s/recommended-repositories. If there are ethical or legal restrictions on sharing a de-identified data set, please explain them in detail (e.g., data contain potentially sensitive information, data are owned by a third-party organization, etc.) and who has imposed them (e.g., an ethics committee). Please also provide contact information for a data access committee, ethics committee, or other institutional body to which data requests may be sent. If data are owned by a third party, please indicate how others may request data access. 8. Please include your tables as part of your main manuscript and remove the individual files. Please note that supplementary tables (should remain/ be uploaded) as separate ""supporting information"" files.

Additional Editor Comments:

When you address the feedback from the reviewers, I kindly ask that you acknowledge the time and effort they have dedicated to your manuscript. Please ensure that your revisions are thorough and that the paper is carefully proofread for content, grammar, and formatting.

Once I receive your revised manuscript, I will evaluate the changes and decide whether to submit it for further review or forward it to the publishers. Please be aware that, depending on the quality of your revisions, there remains a possibility that the referees may still recommend against publication. Therefore, this invitation to revise should not be interpreted as a guarantee of final acceptance.

Reviewers' comments:

Reviewer's Responses to Questions

**Comments to the Author**

1. Is the manuscript technically sound, and do the data support the conclusions?

Reviewer #1: Yes

Reviewer #2: Yes

Reviewer #3: Yes

Reviewer #4: Partly

2. Has the statistical analysis been performed appropriately and rigorously? 

Reviewer #1: N/A

Reviewer #2: No

Reviewer #3: Yes

Reviewer #4: N/A

3. Have the authors made all data underlying the findings in their manuscript fully available?

Reviewer #1: Yes

Reviewer #2: Yes

Reviewer #3: Yes

Reviewer #4: Yes

4. Is the manuscript presented in an intelligible fashion and written in standard English?

Reviewer #1: No

Reviewer #2: No

Reviewer #3: Yes

Reviewer #4: Yes

5. Review Comments to the Author

Reviewer #1: The study entitled "Research on Road Performance and Crack Extension Law of Cement-Phosphogypsum Red Clay under Dry and Wet Cycle" addresses the utilization of cement and phosphogypsum in improving red clay for road construction purposes. However, several critical aspects were not adequately addressed, and the manuscript requires substantial revisions before it can be considered for publication. At this stage, I regret to inform you that I cannot recommend its acceptance.

- Writing Quality: The paper is poorly written and requires significant improvement in terms of language clarity and structure. The current writing lacks coherence, making it difficult for readers to follow the study's objectives, methodology, and conclusions. A thorough revision is needed to enhance the readability and presentation of the research.

- Selection of Tests: The chosen tests, namely California Bearing Ratio (CBR) and Unconfined Compressive Strength (UCS), do not align well with the primary focus of the manuscript. While these tests provide some insight into the mechanical properties, they do not comprehensively address the broader engineering behavior of cement-phosphogypsum-red clay mixtures under dry and wet cycles. The study would benefit from the inclusion of additional relevant tests to assess long-term performance and durability.

- Experimental Design: The number of tests performed, and the selection of parameter values seem inadequate and may not fully represent the material behavior under varying conditions. A more rigorous and detailed experimental plan should be developed to ensure that the conclusions drawn are robust and generalizable.

- Results Presentation: The presentation of results is suboptimal. The data lacks clarity, and the visual representation of findings (e.g., graphs, tables) does not effectively communicate the key outcomes of the study. A more structured and comprehensive approach is needed to present the experimental data in a way that enhances the understanding of the results.

- Discussion and Comparison with Previous Studies: The discussion section is underdeveloped and fails to critically compare the study's findings with relevant literature. A thorough comparison with previous studies is essential to establish the contribution of this research to the existing body of knowledge. The authors should strengthen this section by engaging in a more detailed analysis of how their results align or differ from previous works in the field.

In conclusion, the manuscript, in its current form, requires substantial revisions to address the aforementioned issues. Only after these major revisions are made can the manuscript be reconsidered for publication. At this point, I recommend rejection of the paper.

Reviewer #2: Reviewer#2: Comments to the Author (PONE-D-24-26244)

In this study, the synergistic effect of cement, phosphogypsum and red clay on road performance was studied. The subject of recycling PG in civil engineering applications in recommended. However, the execution falls short. The paper proved challenging to follow, poorly organized, and lacking clarity in its narrative. A major revision should be addressed before the publication of this paper:

1- The abstract section should be improved to highlight the methodology, as well the objectives of the study.

2- Authors needs to precise the novelty of this work in the introduction section

3- The experimental program (i.e., the materials and methods section) is not well-presented. It is difficult for readers to grasp the experimental setups easily. Revision is needed.

4- Please provide a flow chart (picture of specimens) illustrating the experimental procedures.

5- Discussion : Please compare your results with those previously published

You reported data directly without any scientific interpretation. Add a more convincing explanation to support this.

6- Add a discussion section before conclusion regarding practical implementation of current study like the below references:

7- Concerning the practical implications, please compare your findings with existing references for conventional materials like ASTM and RILEM.

8- The environmental feasibility section is not well-presented and requires major revision. It is suggested to adopt Life Cycle Assessment for better clarity.

9- Moderate English changes are required

Reviewer #3: azidane hind

IbnTofail University

B.P. 133, 14000

Kenitra, Morocco

October 03, 2024

Editorial Board of PLOS ONE

Dear Chief Editor of PLOS ONE

Comments to the Author

The manuscript entitled " Research on road performance and crack extension law of cement-phosphogypsumred clay under dry and wet cycle", which submitted to PLOS ONE has been reviewed. The comments are included at the bottom of this letter.

The manuscript has scholarly importance and it seems very interesting. The subject of this manuscript seems good. Please improve the paper writing to look the PLOS ONE paper style.

• Abstract: Complete the methods utilized in this paper to provide a comprehensive overview of the experimental approach.

• Introduction: Clarify whether the global annual production is between 100 to 280 tons or 100 to 280 million tons.

• Section 3: Rewrite the sentence for clarity: "Table 1 presents the basic parameters of the soil samples, while Table 2 illustrates their chemical composition."

• Section 4 Title: Consider renaming the section to "Design of Mix Ratios."

• Section 4: Include a short paragraph highlighting the significance of this technique in enhancing the performance and stability of the mixtures.

• Section 4.1: Provide additional explanation for selecting a 5% value, emphasizing its relevance based on prior research or empirical data.

• Section 4.2: Use shorter sentences to convey information effectively.

• Section 5: Write a brief introductory paragraph summarizing the key objectives and findings of this section.

• Section 5.1: Give some details about curing process

• Section 5.2: Begin this section with a statement on the importance of conducting water stability tests. And rewrite this sentence: At the end of the dry and wet cycle, the specimen was placed in a glass cup containing a certain amount of water, and the water surface did not go over the top of the specimen for about 2~3cm; at regular intervals, photographs were taken to record the changes that occurred in the specimen.

• Section 5.4: Rewrite this section step-by-step with detailed explanations of each process.

• Section 5.6 Title: Revise the title to include all terms followed by their abbreviations. Consider dividing the section into two paragraphs: one for the Scanning Electron Microscopy (SEM) test and another for the X-ray Diffraction (XRD) test.

• Section 5.7: Suggest numbering each step independently to improve clarity and organization.

• Section 6.5: Elaborate on each reaction, detailing the inputs and outputs involved.

• Section 6.6: Compare your results with standard measures for similar roads and discuss the long-term implications of your findings.

• Section 7 Introduction: Start with a concise introductory sentence that encapsulates the main theme of this section.

• Future Analysis: Expand on the rationale for selecting each test and outline your future analysis plans, emphasizing the importance of this research.

The manuscript can be accepted in its current form.

Reviewer #4: The study explores how cement, phosphogypsum, and red clay mixtures perform under dry and wet cycles. Specifically, it investigates unconfined compressive strength, California bearing ratio, water stability, and fissure extension. The research finds that adding SCA-2 water stabilizer improves water stability and resistance to fracture, making the mixture suitable for roadbed fillers.

The study addresses the issue of phosphogypsum accumulation, proposing an engineering solution that combines it with red clay, which could reduce environmental pollution. The findings suggest that the mixture can be used in highway construction, specifically as roadbed filler, aligning the research with practical construction applications.

Nevertheless, some choices and explanations are not clear.

1. The paper could benefit from further explanation regarding the choice of the 5% cement and SCA-2 ratio. Why was this specific proportion chosen, and what justifies this choice for broader applications?

2. More comparisons to similar studies or alternative materials used in roadbed construction would strengthen the argument for adopting this mixture over other options.

3. The results section could provide more clarity through visual aids like more detailed graphs or tables, which would better illustrate trends in compressive strength and CBR values.

4. While the paper mentions the number of wet and dry cycles (up to 5), an assessment of long-term durability beyond this range would make the findings more comprehensive for real-world applications.

The study uses well-known test methods (e.g., unconfined compressive strength tests, water stability tests) following standard protocols. It provides a structured approach to sample preparation and testing under specified conditions.

6. PLOS authors have the option to publish the peer review history of their article (what does this mean?). If published, this will include your full peer review and any attached files.

Reviewer #1: **Yes: **Meysam Bayat

Reviewer #2: No

Reviewer #3: **Yes: **azidane hind

Reviewer #4: No

---

## [Author Response · Author response to Decision Letter 0]

4 Nov 2024

Response to Academic Editorial Editor: We have made changes as suggested.

Response to reviewers:

Reviewer #1:We have made changes as suggested.

Reviewer #2:We have made changes as suggested.

Reviewer #3:We have made changes as suggested.

Reviewer #4:We have made changes as suggested.

---

## [Editor Report · Decision Letter 1]

8 Nov 2024

Road properties of cement-phosphogypsum-red clay under dry and wet cycles

PONE-D-24-26244R1

Dear Dr. Chen,

We’re pleased to inform you that your manuscript has been judged scientifically suitable for publication and will be formally accepted for publication once it meets all outstanding technical requirements.

Kind regards,

Nezha Mejjad, PhD

Academic Editor

PLOS ONE

Additional Editor Comments (optional):

I would like to invite the authors to consider the following minot revision:

- Please make sure that figures and tables shown after the related text and not before the text describing and analyzing them.

- Please revise cerefully the paper english.
---

## [Editor Report · Acceptance letter]

3 Dec 2024

PONE-D-24-26244R1 

PLOS ONE

Dear Dr. Chen, 

I'm pleased to inform you that your manuscript has been deemed suitable for publication in PLOS ONE. Congratulations! Your manuscript is now being handed over to our production team.

Kind regards, 

on behalf of

Dr. Nezha Mejjad 

Academic Editor

PLOS ONE